# A genuine layer 4 in motor cortex with prototypical synaptic circuit connectivity

Naoki Yamawaki[1], Katharine Borges[1], Benjamin A Suter[1], Kenneth D Harris[2,3], Gordon M G Shepherd[1]*

[1]Department of Physiology, Feinberg School of Medicine, Northwestern University, Chicago, United States; [2]Institute of Neurology, University College London, London, United Kingdom; [3]Department of Neuroscience, Physiology, and Pharmacology, University College London, London, United Kingdom

**Abstract** The motor cortex (M1) is classically considered an agranular area, lacking a distinct layer 4 (L4). Here, we tested the idea that M1, despite lacking a cytoarchitecturally visible L4, nevertheless possesses its equivalent in the form of excitatory neurons with input–output circuits like those of the L4 neurons in sensory areas. Consistent with this idea, we found that neurons located in a thin laminar zone at the L3/5A border in the forelimb area of mouse M1 have multiple L4-like synaptic connections: excitatory input from thalamus, largely unidirectional excitatory outputs to L2/3 pyramidal neurons, and relatively weak long-range corticocortical inputs and outputs. M1-L4 neurons were electrophysiologically diverse but morphologically uniform, with pyramidal-type dendritic arbors and locally ramifying axons, including branches extending into L2/3. Our findings therefore identify pyramidal neurons in M1 with the expected prototypical circuit properties of excitatory L4 neurons, and question the traditional assumption that motor cortex lacks this layer.

*For correspondence:
g-shepherd@northwestern.edu

**Competing interests:** The authors declare that no competing interests exist.

**Reviewing editor**: Sacha B Nelson, Brandeis University, United States

## Introduction

'Agranular' cortical regions such as the primary motor cortex (M1; area 4) are so named as they are commonly held to lack layer 4 (L4) (**Brodmann, 1909**). The apparent absence of L4 has strongly influenced theories of cortical organization (**Shipp, 2005**; **Bastos et al., 2012**; **Shipp et al., 2013**). Nevertheless, various observations—such as subtle changes in cell density, expression patterns of various molecular markers, branching patterns of thalamocortical axons, and retrograde labeling termination—suggest that motor cortex might contain some sort of L4 homolog (**Krieg, 1946**; **von Bonin, 1949**; **Caviness, 1975**; **Deschênes et al., 1979**; **Skoglund et al., 1997**; **Cho et al., 2004**; **Kuramoto et al., 2009**; **Rowell et al., 2010**; **Mao et al., 2011**; **Kaneko, 2013**; **García-Cabezas and Barbas, 2014**). For example, *Rorb* expression in mouse S1 is highest in L4 (**Schaeren-Wiemers et al., 1997**) (**Figure 1A**), and a similar if weaker and thinner pattern is seen in M1 (**Figure 1B**), coincident with the L3/5A border (**Schaeren-Wiemers et al., 1997**; **Shepherd, 2009**; **Rowell et al., 2010**). In primate M1, *Rorb* is also expressed but at lower levels than in sensory cortices (**Bernard et al., 2012**), and a recent report presented evidence for the existence of L4 based on cytoarchitecture and SMI-32 labeling patterns (**García-Cabezas and Barbas, 2014**).

Although evidence based on markers is useful and highly suggestive, establishing that M1 truly possesses a functional L4 requires showing that neurons in this band have the same input–output connectivity as their counterparts in sensory areas (**Alfano and Studer, 2012**; **Feldmeyer et al., 2013**). In barrel and other sensory cortices in rodents, L4 is characterized by strong input from primary thalamus, an extensive and largely unidirectional projection to the superficial cortical layers, a paucity of inputs from other cortical areas, and a paucity of long-range cortical outputs (**Petersen, 2007**;

**eLife digest** In 1909, a German scientist called Korbinian Brodmann published the first map of the outer layer of the human brain. After staining neurons with a dye and studying the structures of the cells and how they were organized, he realized that he could divide the cortex into 43 numbered regions.

Most Brodmann areas can be divided into a number of horizontal layers, with layer 1 being closest to the surface of the brain. Neurons in the different layers form distinct sets of connections, and the relative thickness of the layers has implications for the function carried out by that area. It is thought, for example, that the motor cortex does not have a layer 4, which suggests that the neural circuitry that controls movement differs from that in charge of vision, hearing, and other functions.

Yamawaki et al. now challenge this view by providing multiple lines of evidence for the existence of layer 4 in the motor cortex in mice. Neurons at the border between layer 3 and layer 5A in the motor cortex possess many of the same properties as the neurons in layer 4 in sensory cortex. In particular, they receive inputs from a brain region called the thalamus, and send outputs to neurons in layers 2 and 3.

Yamawaki et al. go on to characterize some of the properties of the neurons in the putative layer 4 of the motor cortex, finding that they do not look like the specialized 'stellate' cells that are found in some other areas of the cortex. Instead, they resemble the 'pyramidal' type of neuron that is found in all layers and areas of the cortex.

The discovery that the motor cortex is more similar in its circuit connections to other area of the cortex than previously thought has important implications for our understanding of this region of the brain.

*Svoboda et al., 2010*; *Feldmeyer, 2012*). We found that mouse M1 contains pyramidal neurons in a thin laminar zone at the L3/5A border with all these properties.

## Results

### Thalamocortical (TC) input to M1-L4 neurons

In sensory cortical areas, L4 neurons receive strong thalamocortical (TC) excitation from primary sensory thalamic nuclei (*Douglas and Martin, 2004*; *Feldmeyer, 2012*; *Harris and Mrsic-Flogel, 2013*). If the *Rorb*-expressing zone in M1 is similarly organized, then neurons in that laminar location should receive strong TC input from the primary motor thalamic nuclei, particularly the ventrolateral nucleus (VL). This is suggested by previous anatomical work (*Strick and Sterling, 1974*; *Jones, 1975*; *Cho et al., 2004*; *Kuramoto et al., 2009*; *Kaneko, 2013*); however, while monosynaptic VL input to pyramidal neurons in the upper layers of vibrissal M1 was recently demonstrated using an optogenetic-electrophysiological approach (*Hooks et al., 2013*), it was not possible to determine if this input terminated in a putative L4 or L2/3, as vibrissal M1 is highly compressed due to its location at a cortical flexure (*von Economo, 1929*; *Hooks et al., 2011*).

In this study, we focused on the forelimb area of M1 (*Weiler et al., 2008*; *Tennant et al., 2010*), located in the lateral agranular cortex (area 4) (*Caviness, 1975*) where the upper layers are not compressed in this manner, and putative L4 can be more easily distinguished from more superficial layers. (For convenience, we henceforth refer to this forelimb region simply as 'M1'.) To map input connections, we used an optogenetic strategy (*Hooks et al., 2013*), injecting AAV-ChR2-Venus in VL and subsequently preparing coronal slices containing M1; recording conditions were set to isolate monosynaptic inputs (*Petreanu et al., 2009*).

Laminar profiles of the fluorescence intensity of labeled VL axons showed three peaks, in L1, the L3/5A border, and the L5B/6 border, similar to vibrissal M1 (*Hooks et al., 2013*) (*Figure 2A*). In each slice, we recorded from neurons at the L3/5A border (i.e., putative L4 neurons) and from additional neurons across other layers, thereby obtaining a laminar profile of the excitatory TC input from VL (*Figure 2B,C*). This analysis revealed two distinct peaks of TC input, the uppermost of which coincided with the L3/5A border (normalized cortical depth, ~1/3) (*Figure 2C,D,E*). These data thus indicate that M1 contains neurons in a laminar zone corresponding to L4 that receives strong monosynaptic

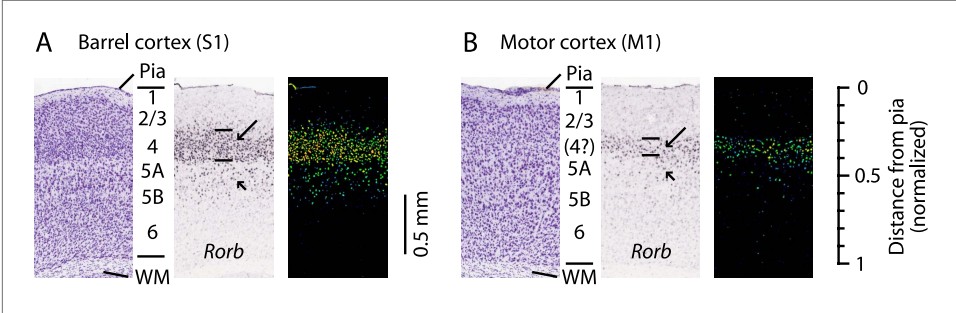

**Figure 1**. L4 in M1 as a zone of *Rorb* expression. Images are from the Allen Mouse Brain Atlas (http://mouse. brain-map.org/) (*Lein et al., 2007*) showing coronal sections for *Rorb* in situ hybridization (probe Rorb-RP_071018_01_H03) and corresponding Nissl stains. (**A**) S1 labeling. In a Nissl-stained section (*left*), L4 is readily identifiable due to cell density differences across layers. In situ hybridization labeling of *Rorb* (*center*, with corresponding expression intensity image shown on the *right*) is the strongest in L4 (long arrow, with borders indicated by lines), with weaker labeling present in L5A/B (short arrow). (**B**) M1 labeling. Nissl stain (*left*) showing a region of the lateral agranular cortex corresponding to the forelimb representation area of M1 (same section as in panel **A**). L4 is not readily identifiable based on cell density differences alone. Nevertheless, in situ hybridization against *Rorb* (*center* and *right*) shows the strongest labeling in a laminar zone corresponding to L4 in S1 (long arrow, with borders indicated by lines), with weaker labeling present in L5A/B (short arrow). Scale on the far right shows the normalized cortical distance from pia to white matter (WM). The approximate location of the cortical layers is indicated, based on prior quantitative analysis of the bright-field optical appearance of M1 layers (*Weiler et al., 2008*).

excitatory TC input from a primary thalamic nucleus associated with this cortical area, thereby fulfilling one important circuit-level criterion for the identification of L4 in M1. For convenience, we henceforth refer to these as M1-L4 neurons.

Prior to investigating the intracortical circuits of M1-L4 neurons (next section), we extended this analysis of TC inputs to address whether M1-L4 neurons also receive inputs from the posterior nucleus (PO) of the thalamus. The fluorescence intensity of the labeled PO axons showed peaks corresponding to L1 and the L3/5A border (but, unlike the VL profile, not the L5B/6 border), similar to the pattern in vibrissal M1 (*Hooks et al., 2013*) (*Figure 3A*). Laminar profiles of the relative amount of monosynaptic TC input from PO axons to M1 neurons (*Figure 3B,C*) showed a broad peak in the upper layers that included L4 and adjacent layers (*Figure 3D,E*). Thus, PO's laminar input pattern in M1 resembled its laminar innervation of secondary somatosensory cortex (S2) (*Pouchelon et al., 2014*) and the septum-related columns of rat barrel cortex (*Lu and Lin, 1993*; *Feldmeyer, 2012*), rather than its innervation of S1 barrel-related columns themselves (*Feldmeyer, 2012*).

## Excitatory output from M1-L4 neurons to L2/3

Next, we tested whether M1-L4 neurons project to L2/3, as L4 neurons do in sensory cortex (*Feldmeyer, 2012*). Previous studies in mouse forelimb M1 using glutamate uncaging and laser scanning photo-stimulation (glu-LSPS) to map local circuits have suggested that neurons around L3/5A border zone can excite L2/3 neurons (*Weiler et al., 2008*; *Wood et al., 2009*; *Wood and Shepherd, 2010*) but lacked the spatial specificity to isolate a putative L4. We examined this pathway by mapping local input to L2/3 neurons at high spatial sampling density (75 μm grid spacing) both in M1 and, for comparison, in the adjacent S1 (*Figure 4A*). To facilitate this side-by-side comparison, in these experiments (only), we used sagittal instead of coronal slices. Synaptic input maps for M1-L2/3 neurons showed a local peak of excitatory input strength arising at the location of the hypothesized L4 (*Figure 4B*). Synaptic input maps for S1-L2/3 neurons were generally similar, but with stronger and spatially more focused excitation from L4, roughly the size and shape of a L4 barrel (*Figure 4C*). The stronger input may partly reflect the higher cell density in barrels (*Figure 1A,B*) (*Hooks et al., 2011*). The laminar profile of L4 input to L2/3 neurons was topographically similar in M1 and S1 (*Figure 4D,E*); that is, the M1 profile was a scaled version of the S1 profile, with a distinct locus of input from L4 (plot in *Figure 4D*). Thus, in M1, the strongest ascending input to the L2/3 neurons arose from the L3/5A border, at a normalized cortical depth of ~1/3, confirming the presence of a L4→2/3 excitatory projection in M1.

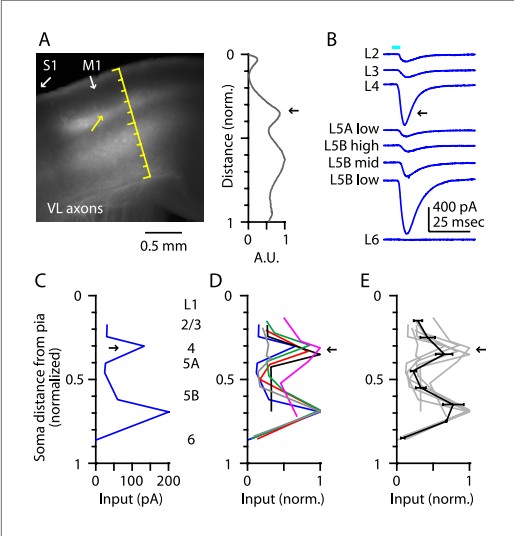

**Figure 2**. Thalamocortical (TC) input to M1-L4 neurons from VL. (**A**) Epifluorescence image of coronal slice containing M1, showing laminar pattern of labeled thalamocortical axons following injection of AAV carrying ChR2 and GFP in the ventrolateral (VL) region of the thalamus. S1 cortex is located laterally (to the left, as indicated). Scale shows normalized cortical distance. Yellow arrow indicates laminar zone of labeling where strong photostimulation-evoked electrophysiological responses were also detected. Plot to the right shows laminar profile of fluorescence intensity, in arbitrary units (A.U.), across layers (normalized distance from pia). (**B**) Responses recorded (sequentially) in vitro in multiple M1 neurons in different layers (as indicated) to photostimulation of ChR2-labeled axons originating from motor thalamus (VL) neurons (**Hooks et al., 2013**). (**C**) Laminar profile of VL input to M1 neurons. The profile exhibits two peaks, one in the upper ~1/3 of the cortex (corresponding to L4) and the other in the lower part (corresponding to L5B). (**D**) Laminar profiles obtained from multiple slices (*n* = 6). Most profiles show a clear peak at a normalized depth of ~1/3 (black arrow). (**E**) Average laminar profile (black; bars: s.e.m.), calculated by binning the data for each profile (bin width: 1/10 of the normalized cortical depth), averaging within each bin, and then averaging across all profiles. The individual profiles are also shown (gray).

A characteristic feature of L4 in sensory cortex is that the projection from L4 to L2/3 is primarily unidirectional, as L4 neurons receive primarily intralaminar excitatory input (**Lorente de Nó, 1949**; **Feldmeyer et al., 2002**; **Schubert et al., 2003**; **Lefort et al., 2009**; **Hooks et al., 2011**; **Feldmeyer, 2012**). To assess whether this also applies to M1, we drew on a previously acquired data set of glu-LSPS input maps (**Weiler et al., 2008**) to further analyze the subset of maps from neurons located in the L4-like zone in M1. The input maps of these neurons typically showed an overall paucity of input, which arose mainly from nearby intralaminar sites (**Figure 5A**). Plotting the median input map of these 10 neurons confirmed that inputs to these L4 neurons arose mostly from L4, with relatively weak input from L2/3 and other layers (**Figure 5B**). Only 2 of 10 neurons received distinct loci of input from other layers: one neuron received relatively weak inputs from L5B/6 (**Figure 5C**) and a second received strong input from L2/3 (**Figure 5D**). These two neurons were at similar laminar locations as the others, suggesting some heterogeneity among M1-L4 neurons' local circuits. Nevertheless, input maps of L4 neurons were generally distinct from those of neurons in mid-L2/3 (see above), and from those of neurons in low-L5A, which typically, and in sharp contrast to the L4 neurons studied here, receive strong L2/3 input (**Weiler et al., 2008**; **Anderson et al., 2010**). Statistical analysis confirmed that L4 input to L2/3 neurons was greater than L2/3 input to L4 neurons by a factor of nearly 4 (L4→2/3: −3.9 pA median amplitude, *n* = 17; L2/3→4: −1.0 pA, *n* = 10; p = 0.0062, rank-sum test) (**Figure 5E**). These analyses thus confirm that L4 neurons in M1 receive mostly intralaminar, rather than interlaminar, excitatory input, and that L4→2/3 projections are predominantly unidirectional.

## M1-L4 neurons receive and send relatively little long-range corticocortical input

L4 neurons in primary sensory areas receive long-range inputs mostly from thalamus but not other cortical areas. In barrel cortex, for example, interhemispheric (callosal) axons from contralateral S1 excite neurons in all layers except L4 (**Petreanu et al., 2007**). We therefore assessed whether M1-L4 neurons similarly receive relatively little long-range cortical input from contralateral M1. We used the same optogenetic-electrophysiological paradigm employed in the TC experiments described above, with AAV-ChR2-Venus injections targeted to contralateral M1. The fluorescence intensity of the labeled corticocallosal axons showed a dip at the location of putative L4 (**Figure 6A**), and laminar profiles of electrophysiologically measured callosal input (**Figure 6B,C**) indicated a nadir at the level of the L3/5A border (**Figure 6C,D,E**). This pattern was complementary to that of thalamic input from VL, that is, callosal input was strong to neurons in L2/3 and weak to those in L4, and vice versa for VL input (**Figure 6F**). Thus, L4 neurons in M1, similar to S1, receive relatively little long-range corticocallosal input from contralateral M1.

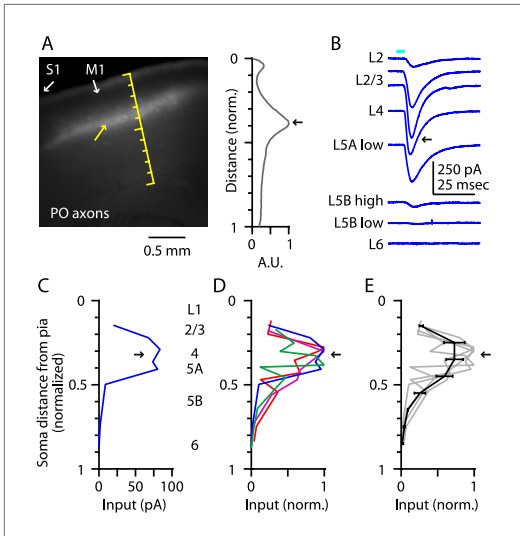

**Figure 3**. Thalamocortical (TC) input to M1-L4 neurons from PO. (**A**) Epifluorescence image of coronal slice containing M1, showing laminar pattern of labeled thalamocortical axons following injection of AAV carrying ChR2 and GFP in the PO region of the thalamus. S1 cortex is located laterally (to the left, as indicated). Scale shows normalized cortical distance. Yellow arrow indicates laminar zone of labeling where strong photostimulation-evoked electrophysiological responses were also detected. Plot to the right shows laminar profile of fluorescence intensity, in arbitrary units (A.U.), across layers (normalized distance from pia). (**B**) Responses recorded (sequentially) in vitro in multiple M1 neurons in different layers (as indicated) to photostimulation of ChR2-labeled axons originating from sensory thalamus (posterior nucleus; PO) neurons (*Hooks et al., 2013*). (**C**) Response amplitudes of the same neurons plotted as a function of laminar location, providing a laminar profile of VL input to M1 neurons. The profile exhibits one peak, situated in the upper ~1/3 of the cortex, somewhat wider (vertically) compared with the peak of VL input, spanning the laminar zone corresponding to L4. (**D**) The laminar profiles obtained from multiple slices (*n* = 4). Profiles show a broad peak at a normalized depth of ~0.2–0.5 (black arrow). (**E**) Average laminar profile (black; bars: s.e.m.), calculated by binning the data for each profile (bin width: 1/10 of the normalized cortical depth), averaging within each bin, and then averaging across all profiles. The individual profiles are also shown (gray).

As well as receiving little long-range input, L4 neurons of sensory cortex typically send only weak long-range corticocortical output. Although exceptions to this rule exist, for example, in primate V1 (*Nassi and Callaway, 2009*), this pattern appears to hold in mouse S1, where a 'gap' is evident in L4 following retrograde tracer injection into ipsilateral vibrissal M1 (e.g., *Sato and Svoboda, 2010*). Furthermore, a similar gap has been noted in the upper layers of vibrissal M1 following retrograde tracer injections in S1, potentially corresponding to L4 (*Mao et al., 2011*). Consistent with this pattern, we observed relatively sparse labeling in M1-L4 following injection of retrograde tracers into contralateral M1 (*Figure 6G*). This laminar gap in labeling was not absolute; higher resolution imaging showed that some of the M1-L4 neurons were clearly labeled with the retrograde tracer but at lower intensity compared with cells in adjacent layers (*Figure 6G*). This gap was centered ~1/3 deep in the cortex (*n* = 3 mice) (*Figure 6H*). Similarly, following injection of retrograde tracers into multiple ipsilateral cortical areas (M2, S1, and S2, in the same animals), we observed a laminar zone with reduced labeling intensity, centered ~1/3 deep in the cortex (*n* = 3 mice) (*Figure 6I,J*). The overall average depth of the local minimum in L4 was 0.35 (normalized cortical depth; ipsi- and contralateral profiles pooled, *n* = 6). These labeling patterns indicate a relative paucity of long-range corticocortical projections originating from L4 of M1, as from L4 of sensory cortices.

## Electrophysiological and morphological properties of M1-L4 neurons

The collective evidence from the preceding experiments indicated that M1 contains neurons having the expected input–output circuits of L4 neurons. Having established that M1 does contain a L4 in the form of these hodologically defined L4 neurons, we next sought to characterize their cellular properties. First, we assessed the electrophysiological properties of these M1-L4 neurons. We recorded 56 neurons located across the upper layers of M1, from upper L2 through L5A. Recordings were targeted to any neurons appearing more likely to be excitatory (pyramidal/stellate) rather than inhibitory based on familiar soma features (shape and size) as observed under bright-field visualization at high magnification (*Hooks et al., 2011*; *Apicella et al., 2012*); all recorded cells had soma features typical of pyramidal neurons, as we did not observe any with stellate-like somata. For each neuron, we analyzed various passive and active membrane properties. Plotting these parameters vs soma depth showed diverse depth-dependent trends and patterns (circles in plots in *Figure 7*). To compare neurons in L4 to those in adjacent layers, we binned the data on the basis of the soma depths into three laminar groups, corresponding to L4, L2/3, and L5A (see 'Materials and methods'). Statistical comparisons (*Figure 7*, *Table 1*) indicated that the

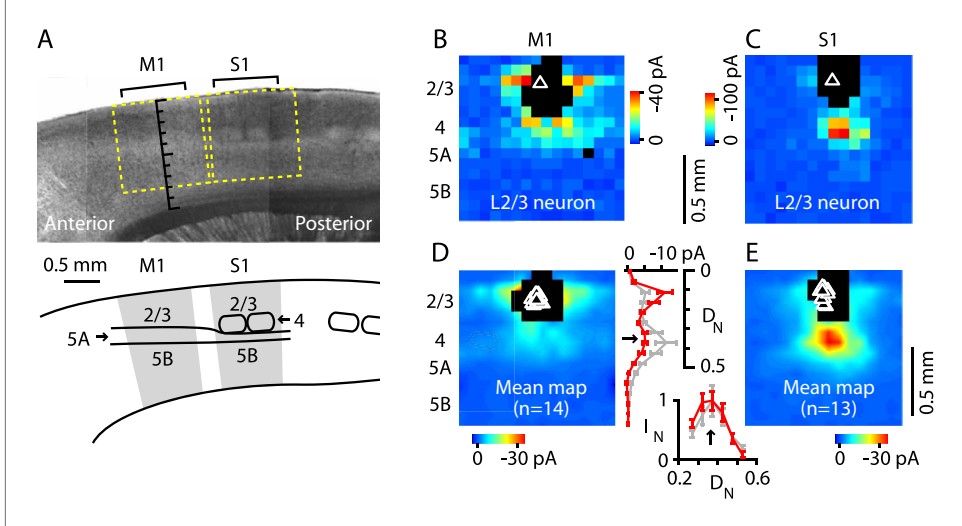

**Figure 4**. Excitatory output from M1-L4 neurons to L2/3. (**A**) Top: bright-field image of a parasagittal slice containing motor (M1) and somatosensory (S1) cortex. In S1, L4 barrels are easily discernable, but are absent from M1, where L5A (lighter-appearing laminar zone) appears wider than in S1. Graduated scale indicates cortical depth in normalized units, from the pia (0) to the white matter (1). Yellow boxes indicate placement of photostimulation grid for mapping inputs to L2/3 neurons in both cortical areas. Bottom: Schematic indicating the major areas and layers of interest in the image. (**B**) Example of a synaptic input map recorded in a L2/3 neuron in M1. Grid spacing was set to 75 μm, the top of the grid was flush with the pial surface, and the grid was horizontally centered over the soma (triangle). Cortical layers indicated to the left, with the location of the L3/5A border (as observed under bright-field) marked by a horizontal line. Inputs arise from both this L4-like laminar zone and the lateral sites in L2/3. Photosimulation sites where the postsynaptic neuron's dendrites were directly stimulated were excluded from analysis and are shown as black pixels. (**C**) Example of a synaptic input map recorded in a L2/3 neuron in S1. Same mapping parameters as in **B**. The input pattern is similar to that of the M1 example shown in B, but with weaker L2/3 and stronger ascending input from the subjacent region corresponding to the L4 barrel layer. (**D**): Mean input map for M1 neurons ($n = 14$). The laminar profile (plotted to the right of the map; red, M1; gray, S1) shows a peak at the level of the L3/5A border (black arrow), ~0.5 mm deep, corresponding to ~1/3 of the normalized cortical depth ($D_N$) in both M1 and S1. The bottom plot shows the L4 region of the same plot, with the input profiles normalized ($I_N$) to their peak values in L4 (arrow); the scaled M1 profile closely resembles the S1 profile. (**E**): Mean input map for S1 neurons ($n = 13$).

electrophysiological properties of L4 neurons were not simply intermediate between those of L2/3 and L5A neurons. For example, L4 neurons were similar to L2/3 neurons but different from L5A neurons in $R_{input}$, $I_{thresh}$, and SFA ratio. Conversely, L4 neurons resembled L5A neurons but differed from L2/3 neurons in $V_r$, AP width, and $V_{thresh} - V_r$. In the case of AP amplitude, the average was greater for L4 neurons than for L2/3 and L5A neurons. L4 neurons showed considerable variability in firing patterns, which ranged from regular and non-adapting to moderately and even strongly adapting, and in other cases highly irregular. From this analysis, we conclude that the properties of L4 excitatory neurons tend to differ from those of either L2/3 or L5A but with considerable cell-to-cell variability, particularly in spiking patterns. Firing pattern diversity has also previously been noted for S1-L4 neurons in rat barrel cortex (**Staiger et al., 2004**).

Lastly, we assessed the morphological properties of M1-L4 neurons. Neurons in slices were filled with biocytin during whole-cell recordings, processed, and imaged with two-photon microscopy (**Figure 8A**). The fluorescently labeled neurons were then digitally reconstructed as three-dimensional tracings ($n = 6$) (**Figure 8B,C**). The morphology of M1-L4 neurons consistently had pyramidal-like dendritic morphology, including a basal arbor with multiple dendrites emerging from the soma, and an apical dendrite extending towards the pia and branching into a small apical tuft in L1 (**Figure 8B,C**). These impressions were borne out by quantitative analysis of dendritic length density across layers (**Figure 8D**). The axonal morphology of these neurons was variable but typically included branches in multiple layers, especially L2/3, L4, and L5A, a pattern evident from inspection of the reconstructions (**Figure 8B,C**) and borne out by length density analysis (**Figure 8E**). This laminar profile of axonal

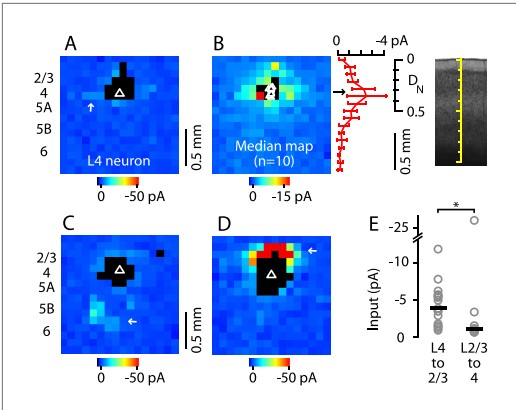

**Figure 5**. Paucity of local input to M1-L4 neurons from L2/3 and other layers. (**A**) Example of a typical synaptic input map recorded from a M1-L4 neuron, showing mostly intralaminar excitatory input (arrow), and little L2/3 input. The triangle marks the location of the soma, and the black pixels represent photostimulation sites resulting in direct dendritic responses. (**B**) Median input map for M1-L4 neurons ($n$ = 10). Excitatory input arose mostly from intralaminar sources, with notably little from L2/3 sites. The laminar profile (plotted to the right of the map; red, median ± median absolute deviation; gray, individual cells) shows a peak at the level of the L3/5A border, ~0.5 mm deep, corresponding to ~1/3 of the normalized cortical depth ($D_N$). (**C**) A M1-L4 neuron showing both intralaminar sources of excitation and ascending excitation from L5B/6 sites (arrow). (**D**) An exceptional M1-L4 neuron. (**E**) Comparison of the individual (gray circles) and median (black lines) amplitudes of L4 input to M1-L2/3 neurons vs L2/3 input to M1-L4 neurons (*p < 0.05, rank-sum test).

density peaked in L4 and steadily declined in the ascending direction across L2/3 towards zero in L1 and also in the descending direction across L5A towards a low baseline level in L5B and L6. The main trunk of each neuron's axon descended towards the white matter (*Figure 8A–C*), a feature that can also be seen in reconstructed L4 axons from the sensory cortex (e.g., *Lübke et al., 2000*; *Staiger et al., 2004*). In most cases, this descending axon could be traced into the white matter, typically coursing laterally, and sometimes but not always sending a branch medially towards the corpus callosum (*Figure 8B*). Overall, this morphological analysis thus indicates that M1-L4 neurons are generally pyramidal neurons, with intracortical axons that ramify mainly in the upper half of the cortex.

## Discussion

We tested the hypothesis that M1, despite lacking a cytoarchitecturally distinct granular layer, nevertheless contains the circuit-level equivalent of L4 in the form of a layer of excitatory neurons at the layer 3/5A border having the same basic synaptic circuit organization as L4 neurons in sensory cortex. Our findings support this hypothesis and additionally reveal area-specific features of these M1-L4 neurons.

The familiar hallmarks of L4 neurons' circuits in sensory areas include (1) input from primary thalamocortical (TC) axons; (2) output to excitatory neurons in other layers, especially L2/3; (3) largely unidirectional L4→2/3 projections (i.e., little input in return from L2/3), and often, although not as a strict rule, (4) a paucity of long-range corticocortical inputs and outputs. Our results provide evidence for each of these features in M1-L4 neurons.

Our results also revealed features that appear distinct from their S1 barrel counterparts. For one, these neurons received TC input not only from VL but also from PO. This contrasts with S1 barrels, where VPM and PO axons target L4 and L5A, respectively (*Feldmeyer, 2012*), but is consistent with previous findings of PO input to neurons in L4 of S2 (*Herkenham, 1980*; *Theyel et al., 2010*; *Pouchelon et al., 2014*) and inter-barrel septa in L4 of rat S1. Another difference from S1-L4 neurons was that the M1-L4 neurons were all pyramidal neurons; we did not detect star pyramids or spiny stellate cells, as are found in rodent S1 (*Staiger et al., 2004*; *Feldmeyer, 2012*). In this respect, M1-L4 is more similar to the primary visual and auditory cortices of rodents, which also do not contain spiny stellate cells (*Peters and Kara, 1985*; *Smith and Populin, 2001*). Indeed, as shown in ferret visual cortex, spiny stellate cells first develop as pyramidal neurons and subsequently lose their apical dendrites through developmental sculpting, indicating a spectrum of L4 morphological subtypes, with 'pyramidal' as the default or prototypical structure (*Callaway and Borrell, 2011*). The finding that all M1-L4 neurons in our sample extended an axon towards and often into the subcortical white matter does not however represent a difference, as this is also commonly observed for S1-L4 neurons (e.g., *Lübke et al., 2000*; *Staiger et al., 2004*; *Shepherd et al., 2005*), and L4 neurons (including stellates) with callosal projections have been described in cat V1 (e.g., *Vercelli et al., 1992*).

In several ways, M1-L4 neurons displayed properties that more closely resembled pyramidal than spiny stellate neurons in rodent S1-L4. For example, the axonal projections of S1-L4 stellate neurons tend to be more focused in a dense beam to L2/3; in contrast, those of S1-L4 pyramidal neurons tend

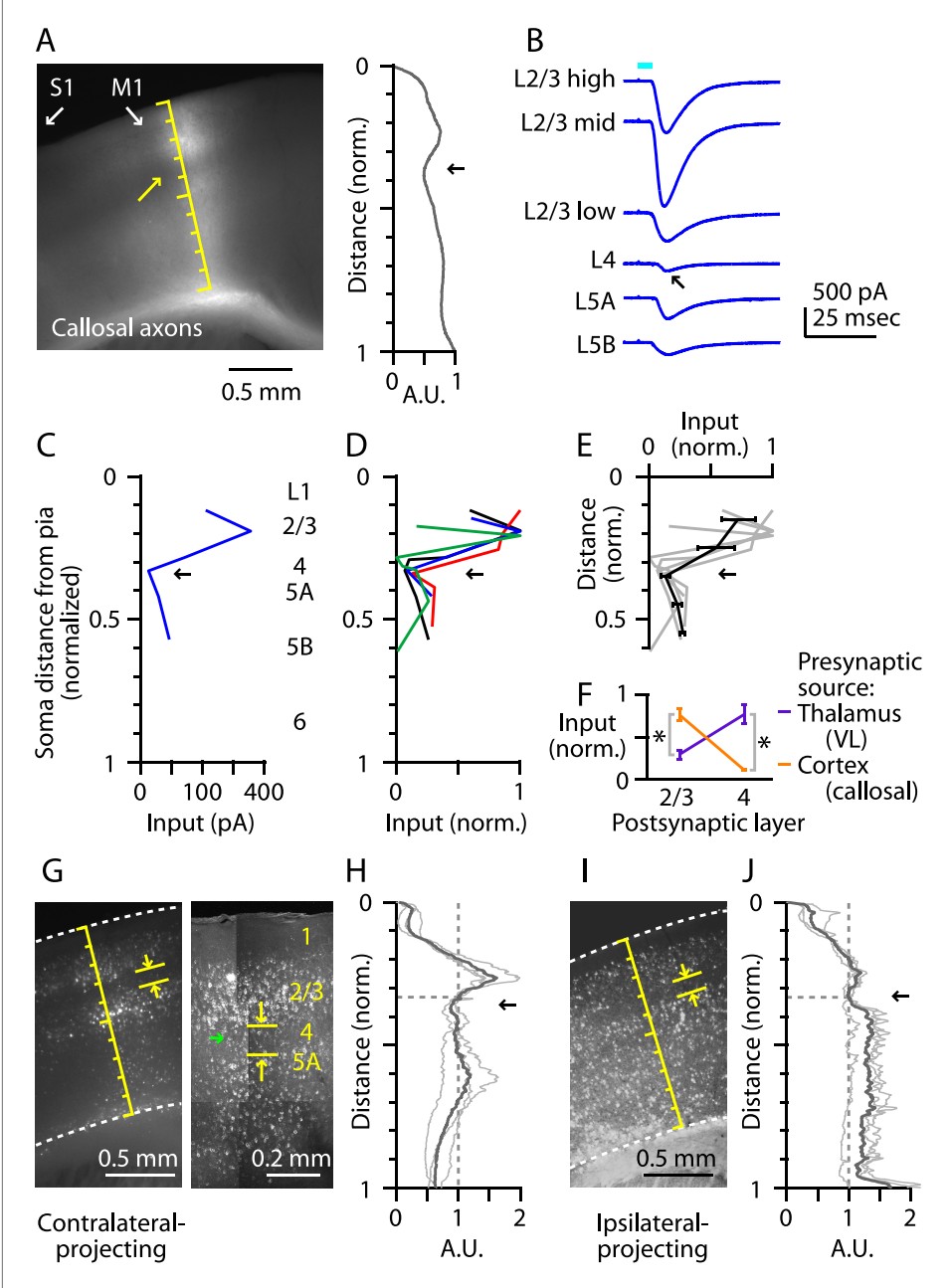

**Figure 6**. M1-L4 neurons receive and send relatively little long-range corticocortical input. (**A**) Epifluorescence image of coronal slice containing M1, showing laminar pattern of labeled interhemispheric corticocortical axons following injection of AAV carrying ChR2 and GFP in the contralateral M1. Scale shows normalized cortical distance. Arrow indicates laminar zone of labeling where weakest photostimulation-evoked electrophysiological responses were also detected. Plot to the right shows laminar profile of fluorescence intensity, in arbitrary units (A.U.), across layers (normalized distance from pia). (**B**) Responses recorded (sequentially) in vitro in multiple M1 neurons in different layers (as indicated) to photostimulation of ChR2-labeled axons of contralateral M1 neurons infected with AAV-ChR2. (**C**) Response amplitudes of the same neurons plotted as a function of laminar location, providing a laminar profile of contralateral M1 input to M1 neurons. The profile exhibits a dip in the upper ~1/3 of the cortex (corresponding to L4) (arrow). (**D**) The laminar profiles were obtained from multiple slices (*n* = 4). Profiles show a dip ~1/3 deep in the cortex (in normalized coordinates). (**E**) Average laminar profile (black; bars: s.e.m.), calculated by binning the data for each profile (bin width: 1/10 of the normalized cortical depth), averaging within each bin, and then averaging across all profiles. The individual profiles are also shown (gray). (**F**) Comparison of input from
*Figure 6. Continued on next page*

*Figure 6. Continued*

callosal axons (from contralateral M1; *n* = 4 slices) vs thalamic axons (from VL; *n* = 6 slices), recorded in postsynaptic L2/3 and L4 neurons in M1 (*p < 0.01, rank-sum test). For each slice, values within each laminar zone (in units of normalized cortical depth: L2/3, 0.1 to 0.25; L4, 0.29 to 0.37) were averaged to obtain a single value per profile; these were averaged and plotted with error bars representing the s.e.m. (**G**) Representative epifluorescence image (left) showing gap (marked by yellow arrows) at the level of L4 (~1/3 deep in the cortex, in normalized distance units) in the retrograde labeling of M1 neurons following injection of retrograde tracers in contralateral M1. Two-photon microscopic image (right) showing the same labeling pattern at a higher resolution (different animal). Some neurons within the 'gap' are labeled with the retrograde tracer (green arrow). (**H**) Laminar profiles of fluorescence intensity of M1 neurons projecting to contralateral M1. Each trace represents the average profile for one animal, obtained by averaging several M1-containing slices. For display, the profiles were normalized to the value in L4 (specifically, the value at a normalized cortical depth of 1/3). The bold line is the average of three animals. There is a reduction in labeling intensity in the L4 region (arrow). (**I**) Representative epifluorescence image showing gap at the level of L4 in the retrograde labeling of M1 neurons following injection of retrograde tracers into ipsilateral M2, S1, and S2. (**J**) Laminar profiles of retrograde labeling pattern for ipsilateral injections. Average traces were calculated as in Panel **G**. The average (bold line) of three animals shows a zone of reduced labeling observed ~1/3 deep in the cortex, corresponding to L4 (arrow).

to show more horizontal spread, lower density in L2/3, and more branching in L5A (*Brecht and Sakmann, 2002*; *Bender et al., 2003*; *Lübke et al., 2003*; *Staiger et al., 2004*; *Feldmeyer, 2012*). Such a pattern is also observed for S1-L4 neurons located in the inter-barrel septa (in rats), which are pyramidal neurons (*Brecht and Sakmann, 2002*; *Bureau et al., 2004*; *Shepherd et al., 2005*). In rat S1, thalamocortical axons from PO branch in L4 in septum-related columns (similar to our finding of PO innervation of M1-L4 neurons) but not in barrel-related columns in rat S1, where they instead branch in L5A (and L1) (*Lu and Lin, 1993*; *Wimmer et al., 2010*; *Feldmeyer, 2012*). The septal region of rat S1 has been proposed to be hodologically organized as a higher order rather than primary sensory cortical area (*Killackey and Sherman, 2003*). Consistent with this, PO projects to L4 in S2 in addition to septal-S1 (*Theyel et al., 2010*; *Pouchelon et al., 2014*). Thus, our findings are generally suggestive that, at least in terms of its L4-related organization, the 'primary' motor cortex more closely resembles higher order than primary sensory cortex.

The apparent absence of L4 in M1 and other agranular cortical areas has long been of interest for its implication that the circuit organization of these areas differs fundamentally from that of sensory areas (*Shipp, 2005*; *Feldmeyer et al., 2013*; *Shipp et al., 2013*; *García-Cabezas and Barbas, 2014*). Our results suggest that L4 in M1 has been 'lost' only at the level of cytoarchitecture but not of cellular connectivity, as it is present in the form of a layer of pyramidal neurons with the expected input–output connections of prototypical L4 neurons. This accords with the general notion that cortical circuit organization tends to be conserved rather than reinvented across areas, with variations arising mostly through quantitative differences in a core set of existing circuits (*Harris and Shepherd, 2015*). For example, rodent M1 possesses not only a thin L4 but an expanded L5B (*Brecht et al., 2004*; *Weiler et al., 2008*; *Yu et al., 2008*; *Anderson et al., 2010*; *Hooks et al., 2013*). In rodent S1 barrel cortex, it is L4 that is instead expanded and shows the most overtly specialized connectivity (*Feldmeyer, 2012*); similarly, L4 in V1 of highly visual mammals is often elaborately differentiated (*Fitzpatrick, 1996*; *Nassi and Callaway, 2009*). Thus, L4 appears to be most elaborate in the primary sensory cortices of modalities that are particularly ethologically relevant to an animal. Motor cortex contains a L4 circuit that is smaller and simpler but retains the same prototypical connectivity patterns. We speculate that like their sensory cortical counterparts, L4 neurons in M1 are specialized for processing of thalamic input before this information is integrated with the activity of other cell classes (which may also be thalamorecipient) downstream in the local M1 network. Our results should facilitate further studies of M1-L4 by enabling a shift of focus away from the question of whether L4 neurons are present in M1 to questions of what types of information they process, how they do so, and how this relates to motor behavior.

## Materials and methods

Animal studies were approved by the Northwestern University Animal Care and Use Committee. In vivo stereotaxic injections of retrograde tracers (fluorescent microspheres, Lumafluor, Durham, NC) or AAV viruses encoding channelrhodopsin-2 (AAV-ChR2-Venus) were performed as described

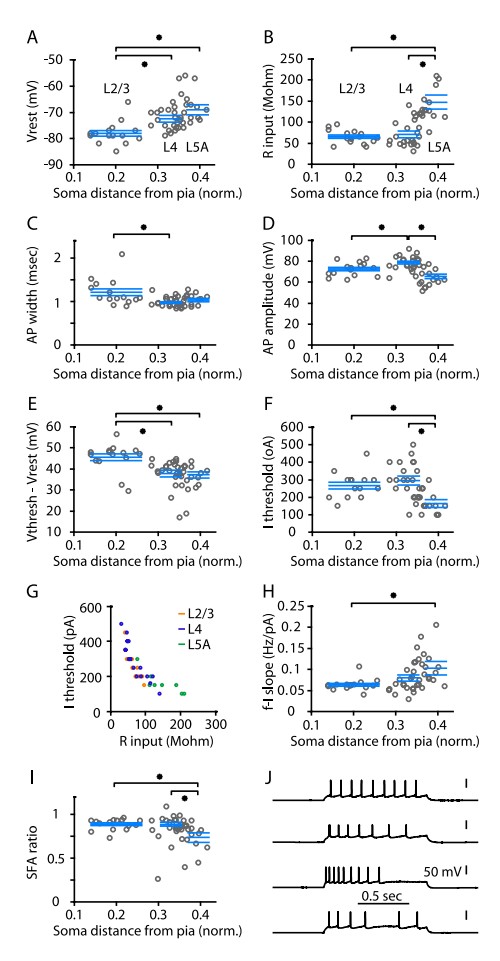

**Figure 7**. Electrophysiological properties of M1-L4 cells. (**A**) Resting membrane potential ($V_r$), plotted as a function of the cortical depth of the soma (normalized distance from pia). A total of 56 neurons were sampled across layers 2/3 through 5A. For analysis, neurons were binned into three main laminar groups corresponding to L2/3, L4, and L5A. The blue lines indicate the laminar range of each group, and also represent their mean ± s.e.m. values. Significant differences between groups are marked (*, rank-sum test). See 'Materials and methods' for additional details. (**B**) Input resistance ($R_{input}$) vs soma depth. (**C**) Action potential (AP) width vs soma depth. (**D**) AP amplitude vs soma depth. (**E**) ($V_{thresh} - V_r$) vs soma depth. (**F**) Current (*I*) threshold vs soma depth. (**G**) $R_{input}$ vs $I_{thresh}$. (**H**) Slope of the frequency–current (F–I) relationship. (**I**) Spike-frequency adaptation (SFA) ratio. (**J**) Example traces, representing the various types of repetitive firing patterns observed among L4 neurons.

(*Anderson et al., 2010*; *Hooks et al., 2013*). Retrograde tracer injections were made in either the contralateral M1 or ipsilateral M2, S1, and S2 of 6- to 7-week-old mice, to label corticocortical projection neurons in M1; brain slices were prepared 3–7 days later and imaged as described below. Viral injections were made in the motor thalamus of 3- to 4-week-old mice, targeting the ventrolateral nucleus, and optogenetic experiments in brain slices were performed ~3 weeks later.

Brain slice preparation and electrophysiology were performed as previously described (*Weiler et al., 2008*; *Anderson et al., 2010*). Whole-cell patch-electrode recordings were made from neurons in 0.3-mm-thick brain slices containing M1. Data were sampled at 10 kHz (most experiments) or 40 kHz (for intrinsic electrophysiology measurements) and filtered at 4 kHz. For optogenetic experiments, recordings in L4 (and other layers) were generally targeted to neurons with 'pyramidal' somata. For experiments aimed at characterizing intrinsic and morphological properties, recordings were targeted to L4 neurons with any soma shape or size except those suggestive of common types of interneurons, particularly basket cells (*Apicella et al., 2012*). Data acquisition was controlled by *Ephus* software (www.ephus.org) (*Suter et al., 2010*).

Standard electrophysiological stimulus protocols were delivered to assess intrinsic properties, as previously described (*Suter et al., 2013*). For each cell, after measuring the resting membrane potential, current was injected as needed to set the membrane potential to −70 mV, and then stimulus protocols were delivered to measure electrophysiological properties. Spike-frequency accommodation (SFA) ratio was calculated as the ratio of the third to fifth inter-spike interval in the first trace containing ≥6 spikes. Current threshold was defined as the amplitude of the current step that was sufficient to evoke one or more action potentials.

For group analyses of electrophysiological properties (*Figure 7*, *Table 1*), statistical comparisons were performed by pooling neurons into laminar groups corresponding to L4, L2/3, and L5A. Based on the results of the circuit analyses (*Figures 1–6*), we defined 'L4' as a thin zone centered on 0.33 (in units of normalized cortical depth) and spanning 0.05 the cortical thickness (i.e., depth range 0.305–0.355). We defined 'L2/3' as the laminar zone spanning 0.14–0.26, and 'L5A' as the zone 0.37–0.42. These laminar zones were separated by small gaps (0.045 between L2/3 and L4, and 0.015 between L4 and L5A), which reduced (but did not necessarily eliminate) the likelihood that some neurons were wrongly classified due to slice-to-slice variability in layer thicknesses. A small number of neurons thus fell outside these groups and were excluded from group analyses (but not from the plots; all data are plotted as circles in *Figure 7*).

**Table 1.** Electrophysiological properties of L2/3, L4, and L5A neurons in M1

| Parameter | L2/3 neurons ($n$ = 15) | L4 neurons ($n$ = 18) | L5A neurons ($n$ = 8) | L2/3 vs L4 | L4 vs L5A | L2/3 vs L5A |
|---|---|---|---|---|---|---|
| $V_r$ (mV) | −78 ± 1 | −72 ± 1 | −69 ± 2 | *0.00079 | 0.089 | *0.00095 |
| $R_{input}$ (Mohm) | 65 ± 4 | 71 ± 7 | 148 ± 17 | 0.99 | *0.00095 | *0.00034 |
| $C_m$ (pF) | 146 ± 12 | 109 ± 11 | 90 ± 7 | 0.038 | 0.42 | *0.0061 |
| Sag (%) | 2.8 ± 0.3 | 4.3 ± 0.6 | 6.0 ± 1.0 | 0.057 | 0.13 | *0.0088 |
| f-I slope (Hz/nA) | 63 ± 4 | 80 ± 8 | 103 ± 16 | 0.12 | 0.19 | *0.0045 |
| $I_{thresh}$ (pA) | 267 ± 19 | 295 ± 26 | 163 ± 23 | 0.45 | *0.0030 | *0.0038 |
| SFA ratio | 0.89 ± 0.02 | 0.89 ± 0.02 | 0.74 ± 0.05 | 0.91 | *0.0054 | *0.0080 |
| $V_{thresh}$ (mV) | −33 ± 1 | −35 ± 1 | −32 ± 1 | 0.19 | 0.21 | 0.85 |
| AP amplitude (mV) | 72 ± 2 | 79 ± 2 | 66 ± 2 | *0.015 | *0.00078 | 0.028 |
| AP width (msec) | 1.21 ± 0.08 | 0.98 ± 0.01 | 1.03 ± 0.02 | *0.0060 | 0.23 | 0.19 |
| $V_{thresh} − V_r$ (mV) | 46 ± 2 | 38 ± 2 | 37 ± 2 | * 0.00028 | 0.32 | * 0.0041 |

Values under each cell group are mean ± s.e.m. Numbers in the last three columns are p-values for comparisons between the indicated groups (rank-sum test; asterisks indicate significant differences). $V_r$, resting membrane potential; $R_{input}$, input resistance; $C_m$, cell capacitance; $I_{thresh}$, current threshold for evoking action potential(s); SFA ratio, spike-frequency accommodation ratio; $V_{thresh}$, voltage threshold for action potential; AP amplitude, action potential peak minus threshold; AP width, action potential duration. See 'Materials and methods' for additional details.

Glutamate uncaging and laser scanning photostimulation (glu-LSPS) were performed as previously described (*Weiler et al., 2008*; *Wood et al., 2009*; *Wood and Shepherd, 2010*; *Shepherd, 2012*), using 3- to 5-week-old mice. As described in 'Results', in one set of experiments, we acquired sets of input maps for L2/3 neurons in M1 or S1; in another set, we further analyzed a subset of glu-LSPS mapping data from a previous study (*Weiler et al., 2008*). Temporal windowing was used to detect photostimulation sites where the postsynaptic neuron's dendrites were directly stimulated (defined as excitatory events arriving within 7 msec post-stimulus) (*Schubert et al., 2001*), and these sites were excluded from analysis (shown in the figures as black pixels).

Optogenetic photostimulation in brain slices was performed as previously described (*Kiritani et al., 2012*; *Hooks et al., 2013*), exploiting the retained photoexcitability of ChR2-expressing long-range axons in slices (*Petreanu et al., 2007*) and using conditions (in particular, tetrodotoxin and 4-aminopyridine in the bath solution) that isolate monosynaptic inputs (*Petreanu et al., 2009*). Responses to blue-LED photostimulation were sampled in voltage-clamp mode for each neuron, and multiple neurons were recorded per slice. Traces were analyzed to determine the average response amplitude in a 50-msec post-stimulus window. For the set of neurons recorded in the same slice, responses were normalized to the strongest response, resulting in a normalized laminar profile for each slice. Profiles from different slices and animals were pooled for group analyses.

Imaging and morphological reconstructions were performed as previously described (*Suter et al., 2013*), by acquiring two-photon image stacks of neurons that had been biocytin filled during slice recordings, fixed, and processed for fluorescent labeling. Three-dimensional reconstructions of axons and dendrites were manually traced (Neurolucida, MBF Bioscience, Williston, VT) and further analyzed using custom Matlab routines (*Source code 1*) to quantify dendritic and axonal length density, as previously described (*Shepherd et al., 2005*).

Images of expression patterns of molecular markers were obtained from the Allen Mouse Brain Atlas (http://mouse.brain-map.org) (*Lein et al., 2007*).

Many analyses involved plotting a parameter of interest as a function of cortical depth, providing a laminar profile of that parameter. To facilitate comparisons across slices, we converted from absolute cortical depth (distance from pia) to a normalized scale, with pia defined as zero and the cortex–white matter border defined as one. To the extent that the thicknesses of individual cortical layers vary as a constant fraction of the total cortical thickness, this normalization procedure

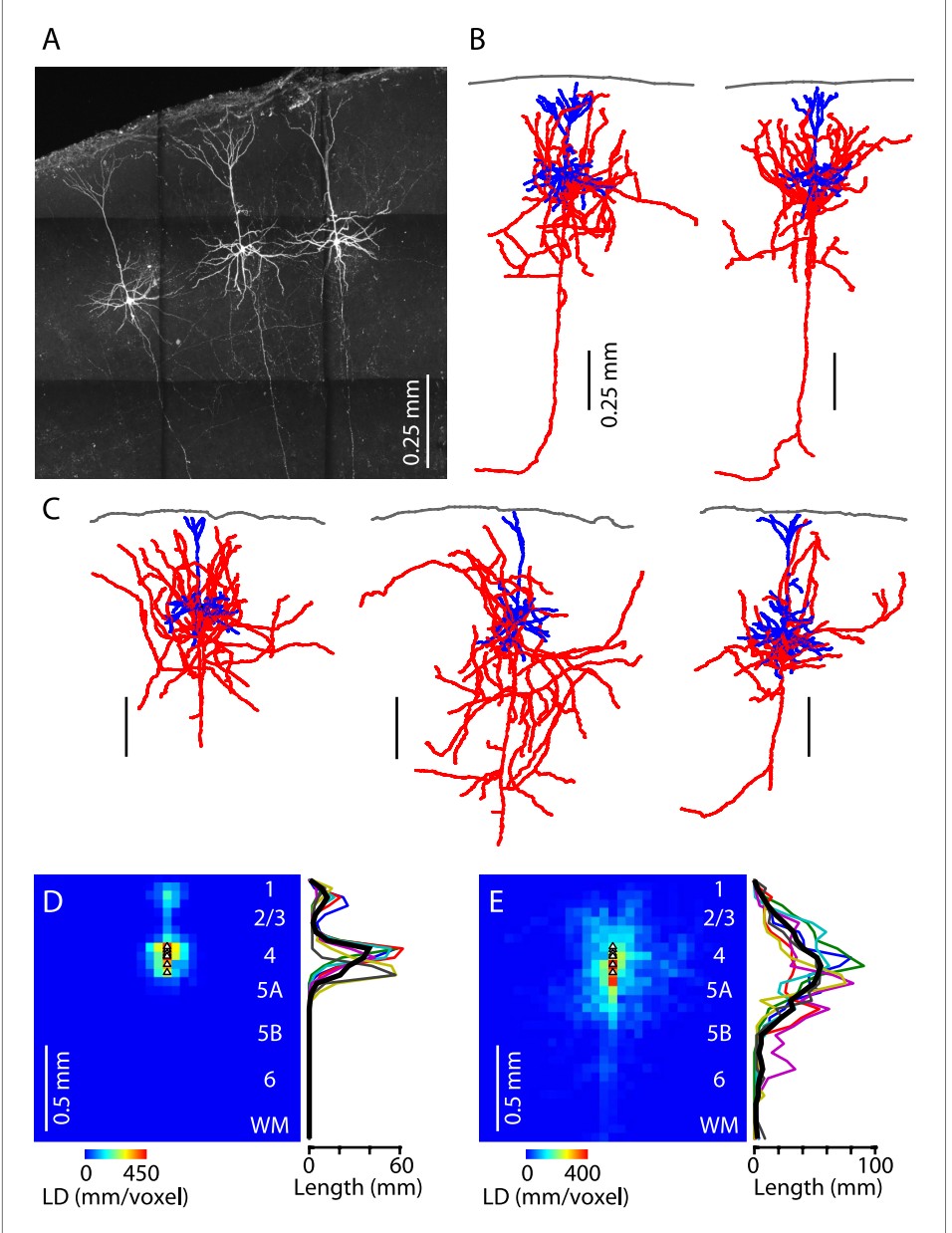

**Figure 8**. Morphological properties of M1-L4 neurons. (**A**) Example fluorescence image of M1-L4 neurons. The neurons were filled with biocytin during whole-cell recordings, fixed and fluorescently labeled, and imaged with a two-photon microscope. The image is a maximum-intensity projection of multiple aligned image stacks. (**B**) Three-dimensional reconstructions of the two L4 neurons in the center and right of the image are shown in panel **A**. Dendrites are blue, axons are red, and the pia is drawn across the top. (**C**) Three more examples. (**D**) Quantitative analysis of dendritic morphology. The three-dimensional digital reconstructions (*n* = 6 neurons) of L4 neurons' dendrites were converted to two-dimensional length–density maps and averaged. Plot to the right shows the same data as a vertical profile (black: group mean ± s.e.m.; colored lines, individual neurons). (**E**) Same analysis, for the axons of the same neurons.

is assumed to reduce some of the slice-to-slice variability; for example, due to small differences in slice angle.

Unless noted otherwise, statistical comparisons were performed using non-parametric tests (rank-sum or signed-rank tests, as appropriate) with significance defined as p < 0.05. For the group analyses shown in *Figure 7* and *Table 1*, significance was defined as p < 0.05/3 (multiple-comparison correction).

## Acknowledgements

We thank D Wokosin for technical expertise and assistance with two-photon imaging, and L Wood and N Weiler for assistance with experiments. Funding support: NIH (NINDS NS061963), Whitehall Foundation, Wellcome Trust.

## Additional information

### Funding

| Funder | Grant reference number | Author |
| --- | --- | --- |
| National Institutes of Health | NS061963 | Gordon M G Shepherd |
| Whitehall Foundation | | Gordon M G Shepherd |
| Wellcome Trust | | Kenneth D Harris |

The funders had no role in study design, data collection and interpretation, or the decision to submit the work for publication.

### Author contributions

NY, KB, BAS, GMGS, Conception and design, Acquisition of data, Analysis and interpretation of data, Drafting or revising the article; KDH, Conception and design, Analysis and interpretation of data, Drafting or revising the article

### Author ORCIDs

Benjamin A Suter, http://orcid.org/0000-0002-9885-6936

### Ethics

Animal experimentation: This study was performed in strict accordance with the recommendations in the Guide for the Care and Use of Laboratory Animals of the National Institutes of Health. All of the animals were handled according to approved institutional animal care and use committee (IACUC) protocols (1248, 1331, 3310) of Northwestern University.

## Additional files

### Supplementary file

• Source code 1. Custom Matlab routines for length density analysis of neuronal morphology.

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
