## [Decision Letter]

Thank you for sending your work entitled “A genuine layer 4 in motor cortex with prototypical synaptic circuit connectivity” for consideration at *eLife*. Your article has been favorably evaluated by Eve Marder (Senior editor) and 2 reviewers, one of whom is a member of our Board of Reviewing Editors.

The following individuals responsible for the peer review of your submission have agreed to reveal their identity: Sacha Nelson (Reviewing editor); Kevan Martin (peer reviewer).

The Reviewing editor and the other reviewers discussed their comments before we reached this decision, and the Reviewing editor has assembled the following comments to help you prepare a revised submission.

Both reviewers thought the experiments performed were valuable and well carried out. One reviewer had mostly minor suggestions for improvement. His only substantive comment centered on the fact that the way that the normalized laminar boundaries was calculated was not adequately justified in terms of slice-to-slice variability. For example in the authors’ prior work (Suter et al.), the SDs of the layer boundaries are given and these are larger than the gaps between the currently assigned layers.

The second reviewer had additional concerns. First, he thought that there was inadequate citation of prior literature. Specifically, he noted “Krieg saw layer 4 in M1 of the albino rat in 1946 and the 1997 analyses of Skogland et al in the rat indicated there are indeed cytoarchitectural features consistent with a layer 4 in M1. Skoglund et al demonstrated quantitatively a middle layer that had higher density and small cells reminiscent of layer 4 in 'granular' cortices. Indeed, their paper is entitled, 'The existence of a layer IV in rat motor cortex'). Cajal reported that human M1 has a layer 4, but thanks to Brodmann it was then was lost to primates and their 'agranular' cortex became a defining feature of M1; see e.g. [52]. Fortunately Garcia-Cabezas and Barbas have just rediscovered layer 4 in the macaque M1.”

Perhaps more importantly, the reviewer was concerned that the case for separating L4 neurons from, for example L3 or L5 neurons was not strong enough and that the criteria were not inadequately specified. Specifically, since the location of the thalamic input has long been known, if the primary criterion is that it is thalamorecipient, this seems like a rather modest advance. On the other hand, one might argue that although the location of the axons was known, the functional strength of input to neurons in different layers that might contact those axons with their basal or apical dendrites was arguably not known.

Finally, the reviewer was concerned about the need for further discussion to indicate whether the authors' prior views of a fundamentally different circuit organization leading to a different laminar sequence of processing for motor cortex (as compared to sensory cortex) is now in need of revision.

---

## [Author Response]

*Both reviewers thought the experiments performed were valuable and well carried out. One reviewer had mostly minor suggestions for improvement. His only substantive comment centered on the fact that the way that the normalized laminar boundaries was calculated was not adequately justified in terms of slice-to-slice variability. For example in the authors’ prior work (Suter et al.), the SDs of the layer boundaries are given and these are larger than the gaps between the currently assigned layers*.

A similar point was raised by reviewer 2. We have addressed this through several changes to the text, including a move of the explanation of the laminar grouping (used for the electrophysiology analyses in Figure 7 and Table 1, where this assignment of cells into layers was done) from the table legend to the Results section, and providing greater detail both in the Methods and Results sections, which should help to clarify these issues.

The SDs in the Suter et al paper presumably overestimate (perhaps substantially) the actual biological variability of the laminar borders, simply because the borders in M1 are rather indistinct and the measurements were made by eye.

*The second reviewer had additional concerns. First, he thought that there was inadequate citation of prior literature. Specifically, he noted “Krieg saw layer 4 in M1 of the albino rat in 1946 and the 1997 analyses of Skogland et al in the rat indicated there are indeed cytoarchitectural features consistent with a layer 4 in M1. Skoglund et al demonstrated quantitatively a middle layer that had higher density and small cells reminiscent of layer 4 in 'granular' cortices. Indeed, their paper is entitled, 'The existence of a layer IV in rat motor cortex'). Cajal reported that human M1 has a layer 4, but thanks to Brodmann it was then was lost to primates and their 'agranular' cortex became a defining feature of M1; see e.g.*
[52]*. Fortunately Garcia-Cabezas and Barbas have just rediscovered layer 4 in the macaque M1*.*”*

We have added a sentence (Introduction, end of first paragraph) regarding primate M1, further emphasizing the (already cited) Garcia-Cabezas and Barbas paper, and also mentioning Rorb labeling in primate M1. We already cited Brodmann, Shipp, and Skoglund (among others), but not the Krieg study (of rat cortex) simply because the Caviness study (of mouse cortex) seemed most relevant; we now cite Krieg as well as this contains an interesting initial description of a cytoarchitecturally apparent L4 in M1. In the same paragraph we also already stated that various observations have suggested the presence of L4 in M1, and cite the Skoglund paper (among numerous others).

The studies mentioned (e.g. Skoglund) primarily address the question of whether L4 is identifiable at the level of cytoarchitecture. Our study addresses the issue at a different level, that of synaptic connectivity (i.e., cellular hodology).

*Perhaps more importantly, the reviewer was concerned that the case for separating L4 neurons from, for example L3 or L5 neurons was not strong enough and that the criteria were not inadequately specified*.

If we understand correctly, this refers primarily to the electrophysiology analyses (Figure 7 and Table 1). The presentation has been modified in several ways to try to address this; please see our response to reviewer 1’s similar comment above.

*Specifically, since the location of the thalamic input has long been known, if the primary criterion is that it is thalamorecipient, this seems like a rather modest advance. On the other hand, one might argue that although the location of the axons was known, the functional strength of input to neurons in different layers that might contact those axons with their basal or apical dendrites was arguably not known*.

Thalamorecipience was not the primary criterion, but rather one of the multiple circuit-level criteria (also including a largely unidirectional projection to the superficial cortical layers; a paucity of inputs from other cortical areas; and a paucity of long-range cortical outputs), which are enumerated the Abstract and elsewhere, and established experimentally in subsequent figures. Regarding the laminar profile of thalamic drive (Figure 2), we were not certain what to expect based on prior literature, and indeed were surprised at the time by what we found. We feel that this evidence, together with our other observations of M1 layer 4’s prototypical connectivity, detailed in later figures, forms a powerful case that at least in mouse, M1 has a genuine layer 4 that may play a similar computational role to layer 4 of other cortices.

*Finally, the reviewer was concerned about the need for further discussion to indicate whether the authors' prior views of a fundamentally different circuit organization leading to a different laminar sequence of processing for motor cortex (as compared to sensory cortex) is now in need of revision*.

We’ve added comments that address this, at the end of the Discussion.